# A scalable workflow to characterize the human exposome

Xin Hu [1], Douglas I. Walker[2], Yongliang Liang[1], Matthew Ryan Smith [1], Michael L. Orr[1], Brian D. Juran[3], Chunyu Ma[4], Karan Uppal[1], Michael Koval [1], Greg S. Martin [1], David C. Neujahr[1], Carmen J. Marsit [5], Young-Mi Go[1], Kurt D. Pennell[6], Gary W. Miller [7], Konstantinos N. Lazaridis [3] & Dean P. Jones [1✉]

Complementing the genome with an understanding of the human exposome is an important challenge for contemporary science and technology. Tens of thousands of chemicals are used in commerce, yet cost for targeted environmental chemical analysis limits surveillance to a few hundred known hazards. To overcome limitations which prevent scaling to thousands of chemicals, we develop a single-step express liquid extraction and gas chromatography high-resolution mass spectrometry analysis to operationalize the human exposome. We show that the workflow supports quantification of environmental chemicals in human plasma (200 μL) and tissue (≤100 mg) samples. The method also provides high resolution, sensitivity and selectivity for exposome epidemiology of mass spectral features without a priori knowledge of chemical identity. The simplicity of the method can facilitate harmonization of environmental biomonitoring between laboratories and enable population level human exposome research with limited sample volume.

[1] Division of Pulmonary, Allergy, Critical Care, and Sleep Medicine, Department of Medicine, School of Medicine at Emory University, Atlanta, GA, USA. [2] Department of Environmental Medicine and Public Health, Icahn School of Medicine at Mount Sinai, New York, NY, USA. [3] Division of Gastroenterology and Hepatology, Mayo Clinic, Rochester, MN, USA. [4] Huck Institute of the Life Sciences, Penn State University, University Park, PA, USA. [5] Department of Environmental Health, Rollins School of Public Health at Emory University, Atlanta, GA, USA. [6] School of Engineering, Brown University, Providence, RI, USA. [7] Department of Environmental Health Sciences, Mailman School of Public Health, Columbia University, New York, NY, USA. ✉email: dpjones@emory.edu

Humans have cumulative lifelong exposure to a million or more commercial, occupational, and environmental chemicals. Forty-seven percent of the 86,405 chemicals registered with the United States Toxic Substances Control Act (TSCA) inventory as of June 2020 are actively manufactured, processed, or imported[1], and each of these has manufacturing impurities and conversion products. Mass spectrometry (MS) provides a powerful chemical analysis platform, and targeted assays are available or possible for almost any chemical. Major, unmet analytical challenges exist for exposome research, however, as a consequence of the number of environmental chemicals and metabolites, chemical diversity, low abundance[2], and lack of readily available authentic standards[3–5]. As a result, few are routinely biomonitored in humans, and many of the commercial chemicals, along with legacy pollutants from prior commercial uses, biotransformation products, and impurities, exist as "dark matter" of the human exposome[2].

An analytical workflow to operationalize untargeted environmental biomonitoring is needed to gain information on known as well as unknown exposures for human exposome research. In contrast to targeted MS analysis, which is developed to measure specific chemicals, untargeted exposome analysis includes measures of known chemicals that are "identified" by MS criteria, and also other MS signals that are unidentified because they have not been associated with known chemicals by MS criteria[6]. These unidentified signals also include chemical contaminants that are known and uncharacterized, as well as reaction products that are effectively unknown to science; the capability to measure these unidentified chemicals in biologic samples is essential to enable population health studies of the dark matter of the exposome.

Procedures, which minimize operator and instrument variation and can be applied consistently for untargeted analysis of tens of thousands of samples, provide a useful approach to address the challenge to measure large numbers of identified as well as unidentified environmental chemicals in human samples. Gas chromatography (GC)–MS is important because many environmental chemicals are hydrophobic, semi-volatile and do not ionize well with popular liquid chromatography (LC)–MS methods. GC–MS is robust, with universally applicable retention-time indices and highly reproducible spectra for database development[7,8]. The high mass accuracy and mass resolution of gas chromatography–high-resolution mass spectrometry (GC–HRMS)[9] in full-scan mode further enhances resolving power to obtain extensive chemical coverage in complex biological matrices. In particular, the high-field Orbitrap detector has sufficient sensitivity and selectivity in full-scan mode and provides the greatest benefit, in terms of ions and compounds monitored as opposed to more targeted acquisitions with single-ion-monitoring (SIM), data-independent acquisitions (DIA), or synchronous SIM/scan[10,11]. Collection of all spectral features enables measurement of known targets based on libraries of authentic standards, while reproducibly measuring and preserving information for unidentified MS features.

With recognition that GC–HRMS in full-scan mode provides reproducibility, extensive coverage, and quantifiable data, key obstacles to implementation of GC–HRMS in exposome epidemiology remain, specifically variability in sample extraction and unautomated data extraction and assembly of the complex data. In LC–HRMS, a single-step sample-extraction procedure[12] improved delivery of data following FAIR principles[13], especially interoperability of data, by eliminating differences due to multistep sample processing. As a result, LC–HRMS is transforming environmental health research through delivery of omics-scale exposure and biologic response data with improved sensitivity, throughput, and affordability[12–14]. In contrast, workflows for targeted GC–MS of environmental chemicals, such as polychlorinated biphenyls (PCBs), polybrominated diphenyl

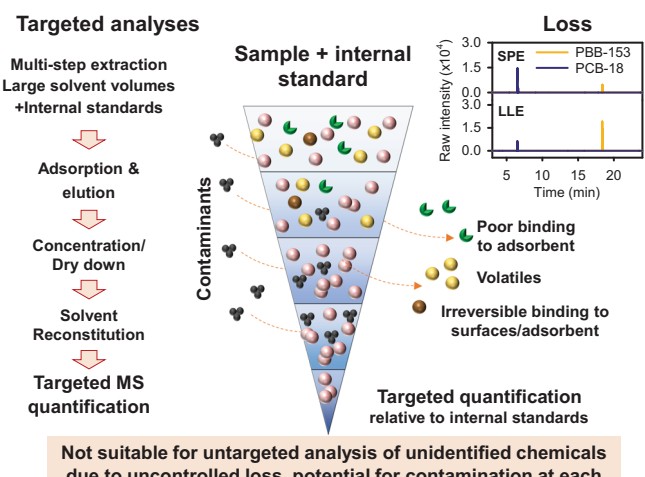

**Fig. 1 A new analytic approach is needed to support health research on extensive and diverse chemical exposures of humans.** Targeted analyses for a limited number of chemicals are available using mass spectrometry. Left: Multistep processing is used for targeted detection of low abundance of environmental chemicals. MS: mass spectrometry. Right: Recovery of chemicals differs by sample-preparation procedures. Standard reference material NIST SRM-1958 was analyzed using solid-phase extraction (SPE) and liquid–liquid extraction (LLE). Middle: Addition of stable isotopic internal standards during the procedure allows correction for variability of recovery due to losses during processing. The multistep processing has potential for contamination at each step, decreases sample throughput, increases cost, and is not suitable for untargeted analysis because there is no way to correct for variable recovery. Source data are provided as a Source Data file for chromatography of SPE and LLE for polybrominated biphenyl (PBB)-153 and polychlorinated biphenyl (PCB)-18.

ethers (PBDEs), polybrominated biphenyls (PBBs), and chlorinated pesticides, use multistep processing to remove biologic matrix effects and enrich the targeted chemicals (Fig. 1)[15,16]. Losses of semivolatile chemicals can occur from dry-down steps following solid-phase extraction (SPE) and liquid–liquid extraction (LLE). Variable loss and contamination can also occur at each processing step. For targeted analyses, inclusion of stable isotopic internal standards overcomes these limitations. For unidentified MS features, however, variability in recovery or loss cannot be evaluated directly (Fig. 1), limiting use for discovery of unidentified environmental chemicals associated with health outcomes.

In this work, we develop a single-step sample-preparation method, which we term express liquid extraction (XLE), for use with GC–HRMS to minimize recovery variability and provide extensive coverage of both identified and unidentified environmental chemicals. We evaluate chemical recovery and used the National Institute of Standards and Technology (NIST) Standard Reference Material-1958 (SRM-1958) to test quantification of environmental chemicals with stable isotopic standards. We establish validity of single-point quantification by reference standardization[17] and show that XLE with GC–HRMS supports quantification of environmental chemicals in diverse human samples, including plasma, lung, thyroid, and stool. We further show a computational workflow that enables untargeted analysis of identified and unidentified environmental chemicals in a form suitable for exposome epidemiology.

## Results

**Sample preparation for GC–HRMS analysis.** Starting initially with a QuEChERS procedure[18,19], we systematically varied

solvent composition, volume, and extraction time (Supplementary Fig. 1) to obtain a simplified procedure with a minimal number of steps and possibilities for contamination and variability in recovery (Fig. 2a). Compared with using dispersive SPE (dSPE) based on the QuEChERS procedure, we found similar reproducibility using high-purity $MgSO_4$ to prepare standard reference material (SRM) of human serum and human plasma samples for GC–HRMS analysis and slightly higher recovery of targeted chemicals using $MgSO_4$ (Supplementary Fig. 2). To avoid contamination by environmental chemicals in solvents and reagents used for QuEChERS, we chose to use high-purity $MgSO_4$. The final extraction procedure, which provided high reproducibility, minimal contamination, high sample throughput, and maximal coverage of chemicals for XLE, used formic acid and hexane:ethyl acetate (2:1) with internal standards, shaking samples in ice-filled cooler on multitube vortexer, centrifugation, and transfer of the organic phase to a new tube, which contained pure $MgSO_4$ to remove water. The results of total ion intensity calculated by the sum of all MS peak intensities showed that the signals in saline extracted by XLE (i.e., method blank) matched the baseline signals found in the directly injected isooctane solvent (Fig. 2b).

**Validation of XLE quantification using standard reference material.** High recovery of [13C]-labeled chemicals was obtained for important classes of environmental chemicals (PCB, PBDE, PAH, and chlorinated pesticides) in NIST SRM-1957 (Fig. 2c). Recoveries ranged from 110 ± 7% for [13C10]mirex to 91 to 105% for congeners of universally [13C]-labeled PCBs, PBDEs, and chlorinated pesticides, with only [13C12]4,4′-dichlorodiphenyldichloroethylene (4,4′-DDE) having low recovery of 65 ± 6% (Fig. 2c). Therefore, the simplified extraction procedure provides an efficient recovery of environment chemicals in an organic phase. Chemicals with less recovery such as 4,4′-DDE can be quantified as long as sample processing is consistent between operators and at different processing times. We evaluated interoperability and found that comparable results (no differences shown by all raw $P > 0.05$, one-way ANOVA) were obtained with the procedure at two different times, seven months apart, and by two different operators (Fig. 2d).

We evaluated quantification using XLE by testing 68 different chemicals (PCB, PBDEs, and chlorinated pesticides) in SRM-1958 using external calibration curves (0.05–2 ng/mL) and comparing measured values with the reference concentrations reported for SRM[20]. We identified all 40 PCBs that are reported with a reference mass fraction (including certified values and noncertified estimates) in the range of 46.6–490 ng/kg in SRM-1958 certificate of analysis (issue date: 11 October 2018). Quantification without adjustment for recovery was reproducible with 29 PCB qualifications at >70% and 35 PCBs at >65% of the reference levels (19 PCBs presented in Fig. 2e, all 40 in Supplementary Data 1). Eleven out of 13 PBDE/PBBs and all 17 organochlorine pesticides were identifiable and reproducibly quantified in this experiment (Fig. 2e and Supplementary Data 1). Therefore, XLE provides sufficient recovery to support accurate absolute quantification of a broad range of environmental chemicals. Overall, XLE supported measurement of 68 out of the 70 chemicals that are in the ng/kg range in SRM-1958 (Fig. 2f and Supplementary Fig. 3).

In addition, we tested XLE quantification of chemicals in a nonfortified reference material: SRM-1957. The results show that XLE detected 29 out of 32 chemicals with certified or estimated reference values in the ng/kg range in SRM-1957, 16 out of 29 chemicals were quantified at >65% of the reference levels[21] (Supplementary Data 1).

**Reference standardization for XLE-based exposome analysis.** Absolute quantification of chemicals in human samples is often complicated by ion-suppression effects of the biologic matrix on chemical detection by mass spectrometry. Stable isotope dilution addresses both recovery issues described above and matrix effects on ionization efficiency and is therefore ideal for quantification in targeted chemical analysis. Use of stable isotopic standards is not practical for untargeted analysis of large numbers of environmental chemicals or for unidentified environmental chemicals. For LC–HRMS analyses, single-point quantification by reference standardization has been established as a useful alternative. To provide this utility for GC–HRMS (Fig. 3a), we tested the ability of reference standardization[17] as a simple and practical approach in untargeted exposomics to estimate chemical concentrations using a single-point calibration. We analyzed 20 human plasma samples and performed reference standardization of 17 chemicals based on SRM-1958 that were processed in parallel to the plasma samples. The selected chemicals included six PCBs, seven chlorinated pesticides, and four PBDEs that are detected in human biomonitoring studies[22,23]. The chemicals were quantified by external standard curves (ranging from 0.05–2 ng/mL) and reference standardization (Fig. 3b). Among the 20 plasma samples, the measured concentrations from two quantification methods were similar (Fig. 3b), with $|\rho| = 1.00$ (Spearman's correlation) for all 17 chemicals (Supplementary Data 2). Additionally, we showed consistent quantification for six of the most prevalent chemicals in human lung samples ($n = 11$), comparing reference standardization to response factor (RF)-based quantification with RF determined by spiked [13C] internal standards[24] (Fig. 3c and Supplementary Data 3). Thus, the results validate use of reference standardization as a simple approach for quantitative analysis of human plasma and tissue in a high-throughput XLE workflow with GC–HRMS.

**Application of the XLE workflow for analysis of environmental chemicals in human samples.** We analyzed 80 archival samples from individuals (57 females, 23 males, aged 41–68 years) without known disease or occupational or environmental exposures of concern as a pilot to test the utility of XLE in large-scale human biomonitoring studies. For targeted environmental chemical analysis, we validated the linearity of 378 chemicals over a concentration range that was relevant for general population analyses (0.05–2 ng/mL) and selected these as our major database for targeted analysis (Supplementary Data 4). Using a requirement for at least three co-eluting accurate mass $m/z$ features (±5 ppm) within 30 s of database-retention time, we identified 49 chemicals belonging to various environmental chemical classes. An unsupervised 2-way hierarchical cluster analysis (HCA) of log-transformed intensity showed clustering according to chemical class (Fig. 4a). In particular, persistent chemicals were highly correlated with each other (all raw $P < 0.001$), including 4,4′-DDE, PCBs 153, 180, 138, 118, and 74, PBDE-47, hexachlorobenzene (HCB), and trans-nonachlor (Fig. 4b). The results showed a general increase of chemical levels with increasing age quartiles (Q3 and Q4: 53–68 versus Q1 and Q2: 41–52) using unsupervised clustering, a trend particularly evident for the cluster of 4,4′-DDE, PCBs 153, 180, 138, 118, and 74, PBDE-47, HCB, and trans-nonachlor. Examination of data according to body mass index (BMI) showed that individuals with BMI ≥ 40 had lower levels of environmental chemicals, which may be attributed to high lipophilicity and propensity to distribute in adipose tissue versus plasma (Fig. 4a). Quantification with reference standardization (Supplementary Data 5) showed that use of two SRM samples with differing environmental chemical concentrations can overcome variable batch effects in

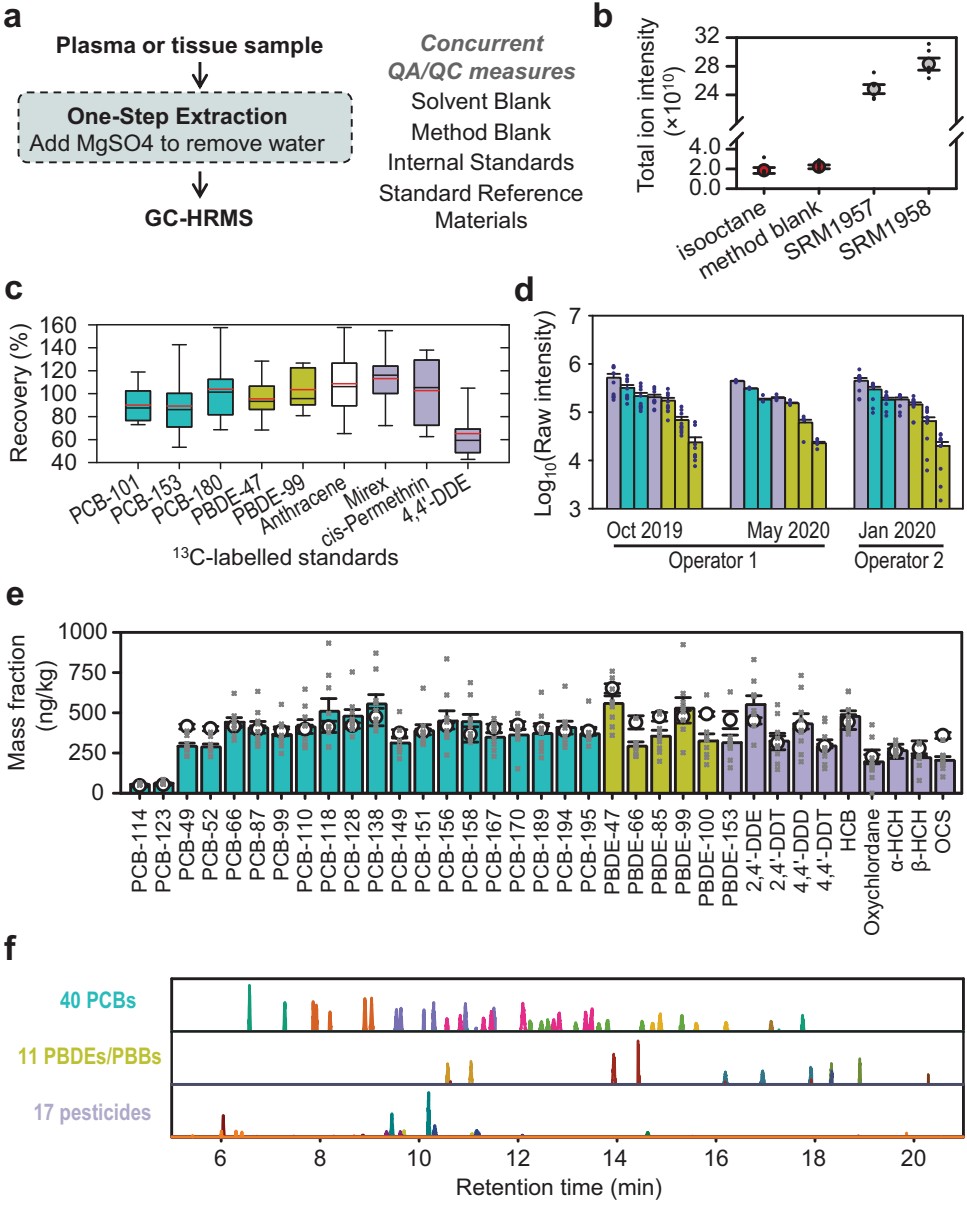

**Fig. 2 XLE provides single-step extraction to minimize variation in chemical recovery for untargeted environmental biomonitoring. a** XLE minimizes the number of sample-processing steps for use with GC–HRMS for measurement. **b** Inclusion of method blanks showing similar signal level as directly injected isooctane solvent supports quality control (QC) and standard reference material (NIST SRM-1957, 1958) in each sample batch supports quality assurance (QA). Total ion intensity was calculated by the sum of intensity of all detected peaks ($n = 6$ independent experiments). Data are presented as average ± SEM. **c** XLE supports recovery of stable isotopic standards calculated as percent recovered in XLE-processed SRM-1958 ($n = 16$ independent experiments) as compared with unprocessed hexane and ethyl acetate mixture ($n = 9$ independent experiments). Red and black line: mean and median, respectively, whiskers: 10th and 90th percentile. **d** XLE is reproducible by independent operators at different analysis time. Selected chemicals that are commonly detected in human biomonitoring studies[22] are presented in raw intensity (average ± SEM, $n = 4$ independent experiments for Operator 1, May, $n = 10$ for Operator 1, October and Operator 2) and illustrate comparability of results as well as long-term robustness. Pesticides are colored in purple, with polychlorinated biphenyls (PCBs) in teal, and polybrominated diphenyl ethers (PBDEs) and polybrominated biphenyl (PBB) in yellow-green. Left-to-right chemicals: 4,4′-Dichlorodiphenyldichloroethylene (DDE), PCB-138, PCB-118, hexachlorobenzene (HCB), PBDE-47, PBDE-153, and PBB-153. **e** Quantification of selected chemicals in SRM-1958 (average ± SEM, $n = 9$ independent experiments) using external standard curves (0.05–2 ng/mL in hexane and ethyl acetate mixture). Reference values of chemical mass fraction are presented by open circle (average ± SEM[20]). Because unidentified environmental chemicals will not have recovery data, these values are expressed without correction for recovery to illustrate usefulness for untargeted analysis (e.g., compare the height of a bar to an open circle). DDD dichlorodiphenyldichloroethane, DDT dichlorodiphenyltrichloroethane, HCH hexachlorocyclohexane, OCS octachlorostyrene. **f** Chemical chromatography in XLE-analyzed SRM-1958 shows detection of an extensive number of environmental chemicals for quantification (peaks are color-coded by primary $m/z$, with chemical details in Supplementary Fig. 4). Source data are provided as a Source Data file.

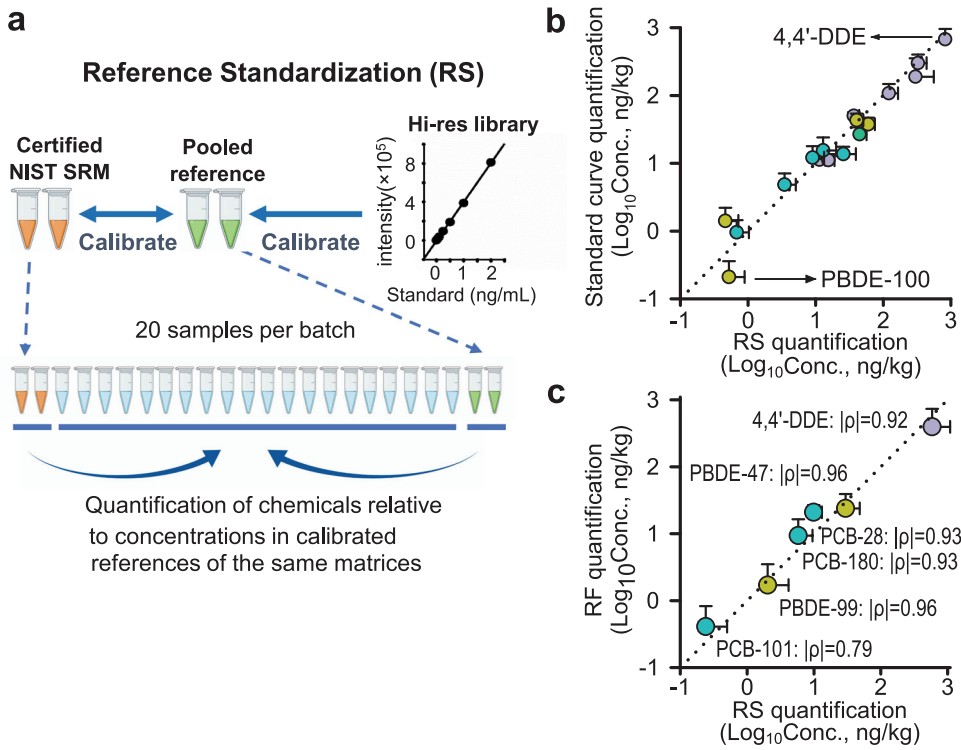

**Fig. 3 Adaptation of reference standardization to GC–HRMS provides automated workflow for quantification of identified environmental chemicals.**
**a** Reference standardization (RS) is a validated single-point quantification method used in liquid chromatography–high-resolution mass spectrometry (LC–HRMS) for automated measurement in high-throughput analyses. The method relies upon pooled reference materials that are calibrated relative to external calibration curves and relative to other pooled reference materials. These pooled reference materials are analyzed before and after each batch of samples, so that concentrations of environmental chemicals detected in the samples can be quantified relative to concentrations in the pooled reference materials, which may be validated by external calibration with high-resolution (Hi-res) chemical library. **b** We used dilution series of environmental chemical standards to validate quantification by reference standardization; mean concentrations of plasma samples ($n = 20$ biologically independent samples) are presented with positive SEM by two methods. **c** We determined response factors (RF) using stable isotopes to validate quantification by reference standardization; mean concentrations of lung samples ($n = 11$ biologically independent samples) are presented with positive SEM by two methods. Spearman's correlation $|\rho|$ was presented. DDE dichlorodiphenyldichloroethylene, PBDE polybrominated diphenyl ethers, PCB polychlorinated biphenyls. Pesticides are colored in purple, with polychlorinated biphenyls (PCBs) in teal, and polybrominated diphenyl ethers (PBDEs) and polybrominated biphenyl (PBB) in yellow-green. Source data are provided as a Source Data file.

quantification for large-scale studies. Examples of the most frequently detected chemicals show that overall distributions were positively skewed by a small subset of individuals with high concentrations (Fig. 4c).

Since only halogenated compounds have reference values reported in SRM-1957 and SRM-1958, we specifically tested detection of nonhalogenated compounds using a 305-chemical library covering a broad range of chemicals (Supplementary Data 6). The results showed detection and quantitative intensity of 33 chemicals, including PAHs, insecticides/pesticides, flame retardants, plasticizers, flavorants and food additives, phthalates, and chemicals used for personal care in human plasma (Supplementary Data 7), thereby establishing usefulness for diverse environmental chemicals.

We tested the general utility of XLE in a variety of human biological samples by analyzing human lung and thyroid tissues and stool samples. We quantified 32 environmental chemicals in 11 human lungs, with HCB, PCB-28, and PCB-18 being most frequently detected (10 out of 11) (Supplementary Data 8). The commonly detected chemicals in human plasma were detected less frequently in the lung. For the 11 lungs, 4,4′-DDE was detected in eight, PCB-153 in five, PBDE-47 and PCB-138 in four, and PCB-180 in three. Although the plasma samples were from nondiseased individuals and the lungs were from both diseased and nondiseased individuals, HCA results suggest that

environmental chemical profiles in human lung may be very different from plasma. Indeed, PCB-18, PCB-28, and HCB, which are relatively volatile, were among the most abundant chemicals in the lung (Supplementary Data 8) in contrast to the prevalence of less volatile chemicals in the plasma (Supplementary Data 5), indicating a potential contribution of inhalation exposure to the more volatile environmental contaminants.

In the small number of thyroids that was analyzed with XLE, 14 environmental chemicals were quantified (Supplementary Data 9). The most prevalent was 4,4′-DDE, detected in four out of five thyroid samples, with a median concentration of 2.20 ng/g. The amounts of individual chemicals were highly variable among the individuals, and the small number of samples precludes any generalization. Nevertheless, HCA of the correlation matrix that showed high correlation of chemicals measured in the thyroid samples was similar to that in the lung and plasma (Fig. 4b).

Human stool samples, as a noninvasive matrix, have unique value in exposome research[25,26] but have not been extensively studied for environmental chemical exposures. For lipophilic and unabsorbed dietary environmental chemicals, stool is a primary route of elimination[26] and can therefore provide useful information on body burden and clearance of chemicals[25]. In a pilot analysis of six human stool samples, we detected 52 and quantified 21 environmental chemicals, with HCB found in all samples (Supplementary Data 10). Quantification of HCB showed

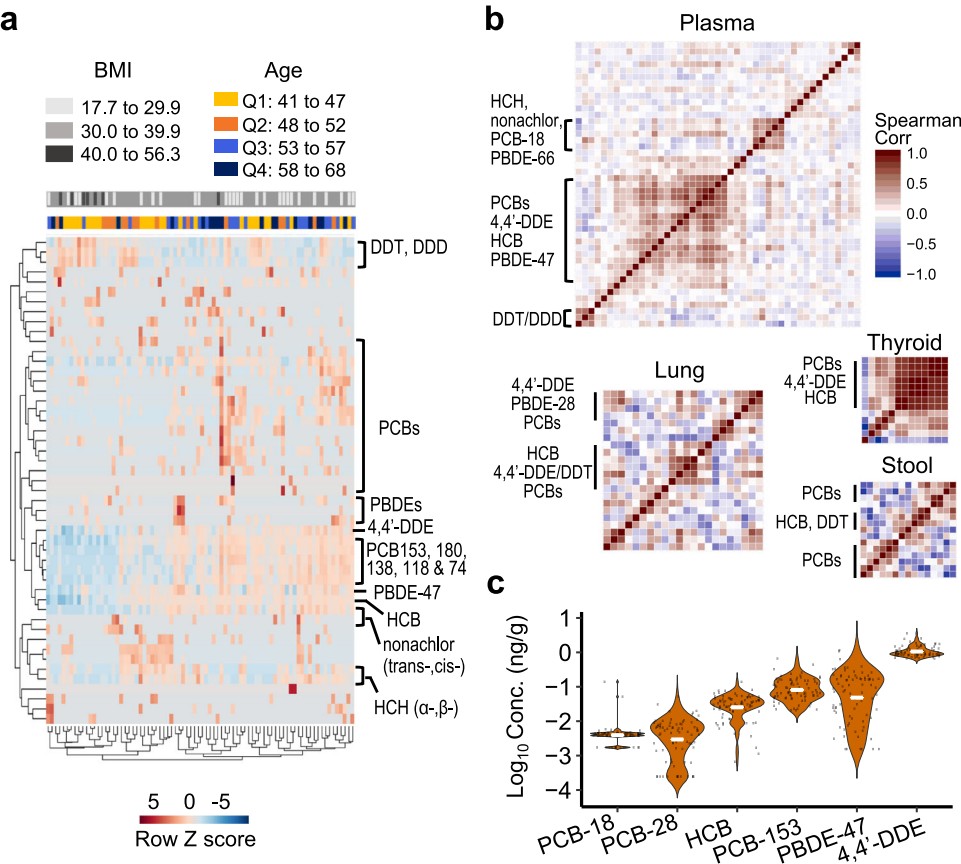

**Fig. 4 XLE with GC–HRMS supports detection and quantification of environmental chemicals in different biologic materials. a** Human plasma from 80 individuals without known occupational or environmental exposures of concern was analyzed for 49 chemicals and visualized by unsupervised two-way hierarchical clustering of log-transformed intensities. Subjects were color-coded into three body mass index (BMI) groups and age quantiles (Q1–Q4). The results show that high-throughput analysis of environmental chemicals enables study of relationships of chemical distributions and associations with health characteristics. **b** XLE supports quantification of environmental chemicals in human plasma ($n = 80$ biologically independent samples), lung ($n = 11$ biologically independent samples), thyroid ($n = 5$ biologically independent samples), and stool ($n = 6$ biologically independent samples) samples. Hierarchical cluster analysis of Spearman's correlation among quantified environmental chemicals shows co-exposure of different chemical classes and illustrates that XLE supports quantification of multiple classes of chemicals in different tissue types for integrative analyses of diverse chemical exposures. **c** Concentrations and distribution of chemicals prevalent in plasma ($n = 80$ biologically independent samples) samples shows that XLE is suitable to support quantification in human plasma (white bar—median; black dot—individual sample concentration). DDD dichlorodiphenyldichloroethane, DDT dichlorodiphenyltrichloroethane, DDE dichlorodiphenyldichloroethylene, HCB hexachlorobenzene, HCH hexachlorocyclohexane, PBDE polybrominated diphenyl ethers, PCB polychlorinated biphenyls. Source data are provided as a Source Data file.

a median concentration of 0.143 ng/g. HCA of the correlation matrix showed that co-exposures of chemicals are likely as shown in the plasma, lung, and thyroid (Fig. 4b). The high correlations of these persistent chemicals are not surprising as they likely derive from similar environmental exposure events. Along with the pilot studies of plasma, lung, and thyroid, these results establish feasibility to use XLE with GC–HRMS for high-throughput quantification of environmental chemicals in biomonitoring.

**Analysis of unidentified environmental chemicals.** An important goal of untargeted biomonitoring is to develop procedures to study the unknown exposures of the human exposome. Suspect screening with chemical databases enables collection of information on known chemicals but not on other chemicals, contaminants, and transformation products. In studies of LC–HRMS of human plasma, half of the *m/z* features associated with health outcomes are unidentified[27]. In principle, statistical and bioinformatics analyses can be performed on accurate MS features obtained from GC–HRMS without chemical identification, and these features can then be used with index chemicals to define

retention to obtain characterization for identification by database searching or deposition into data libraries (Fig. 5a).

We tested the feasibility of XLE with GC–HRMS to capture information on unidentified *m/z* features in a usable form for entry into a data library. We selected all features within a 2-min interval (18.00–20.00 min) detected from the human plasma analyses (4747 *m/z* features) and human lung analyses (5415 *m/z* features) and performed unsupervised spectral clustering with RAMclustR[28]. The results provided 2413 features in the plasma aligned into 341 spectral clusters, and 4041 features in the lung aligned into 573 spectral clusters, each potentially representing an unidentified chemical (Fig. 5b). The rest of the features did not form any clusters that contained three or more features. The majority of the spectral clusters were unidentified based on library search against the NIST/EPA/NIH Main Library containing more than 100,000 compound MS spectra (Xcalibur v 4.2.1).

Selected examples of spectral clusters, identified at the same retention time in both plasma and lung, showed potential to use accurate MS signal to aid in chemical identification via computational tools and fragmentation database (Fig. 5b). Using MS-Finder (ver 3.42)[29,30], we predicted that P106 (Plasma cluster

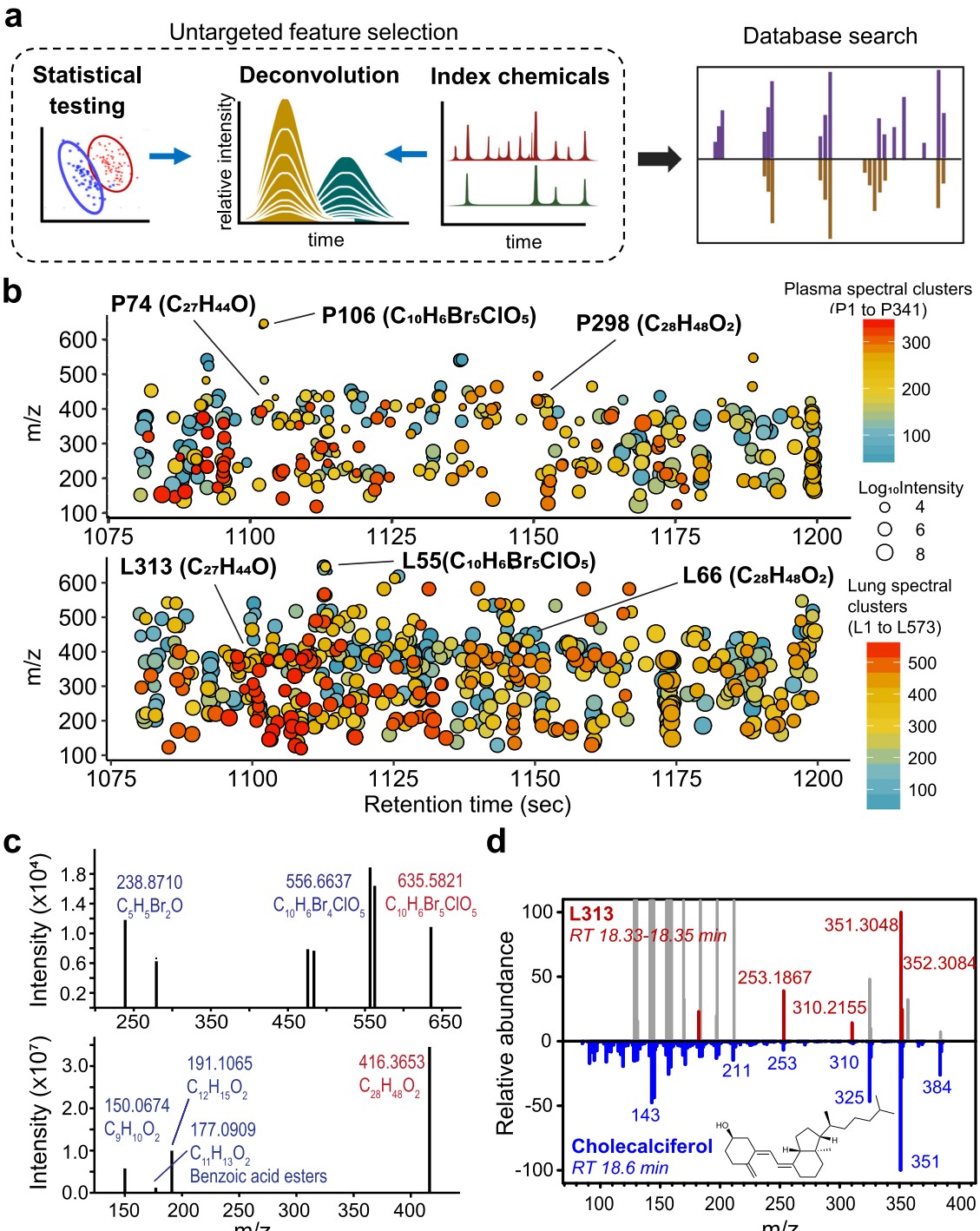

106) may contain a series of *m/z* features with accurate mass matches to halogenated hydrocarbons containing Br and Cl, and P208 may contain a series of hydrocarbons matching elemental compositions containing two oxygens (Fig. 5c). NIST/EPA/NIH Main Library search of L313 (Lung cluster 313) showed 98% probability of L313 being cholecalciferol based on spectrum match and retention index (Fig. 5d). Matches to library search that were missing in L313 spectra were not clustered into L313 by RAMClustR due to lower correlations with L313 spectra across all samples, which may result from possible matrix interference, especially in low *m/z* range. Further confirmation was provided by matching the retention time to authentic standard (Supplementary Fig. 4). These results verify that XLE can be used with

GC–HRMS to perform untargeted analyses of human biospecimens, capturing chemical information that can be later analyzed and used for identification and supporting this untargeted approach for cumulative libraries in exposome research.

## Discussion

XLE with GC–HRMS provides a high-resolution exposomics workflow to address a critical need for public health research, namely, an affordable, interoperable method for population research to study health effects of an extensive number of low-abundance chemicals to which humans are exposed in a single biological sample. As much as 85% of chronic disease is

**Fig. 5 XLE with GC–HRMS measures unidentified environmental chemicals to support exposome epidemiology. a** Data from untargeted analyses can be used directly for biostatistical and bioinformatics analyses of relationships to health markers without chemical identification. For instance, in untargeted feature selection, principal component (PC) analysis can select chemical features that discriminate sample groups (left: red and blue), while chromatographic retention index relative to known index chemicals (right: dark green), can inform properties of selected features. Deconvolution defines the accurate mass spectral signals (middle: golden), which along with retention time and ion intensity, can be incorporated into exposome reference databases and used for subsequent investigation, such as database searches (purple: library spectra; golden: experimental spectra). **b** Application of tools such as RAMclustR[28] to untargeted data allows co-eluting *m/z* features to be studied as possible products derived from an unidentified chemical. In this example of an analysis of human plasma (*n* = 60 biologically independent samples) and lung (*n* = 11 biologically independent samples), unidentified signals of a two-minute retention-time interval are clustered into spectra and color-coded based on clusters. Size of circles reflects weighted intensity calculated based on compiled spectra[28]. **c** Clustered *m/z* spectra are likely to include unidentified environmental chemicals and can be used for discovery of unidentified chemical structures. Examples are presented showing putative molecule formula assigned to spectra by MS-Finder ver 3.42[30]. Candidates with the highest formula scores (P106: 2.4; P208: 4.3) in MS-Finder were selected. Blue—putative fragment ion; Red—putative precursor ion. **d** One spectral cluster (L313: red) was confirmed by library search of spectra (blue) and retention-time match with authentic standard (Supplementary Fig. 4) showing that XLE supports identification of chemicals using an untargeted approach. Missing spectral matches were identified at the same retention time (gray) and were not clustered with L313 by RAMclustR due to low-intensity correlations. Source data are provided as a Source Data file.

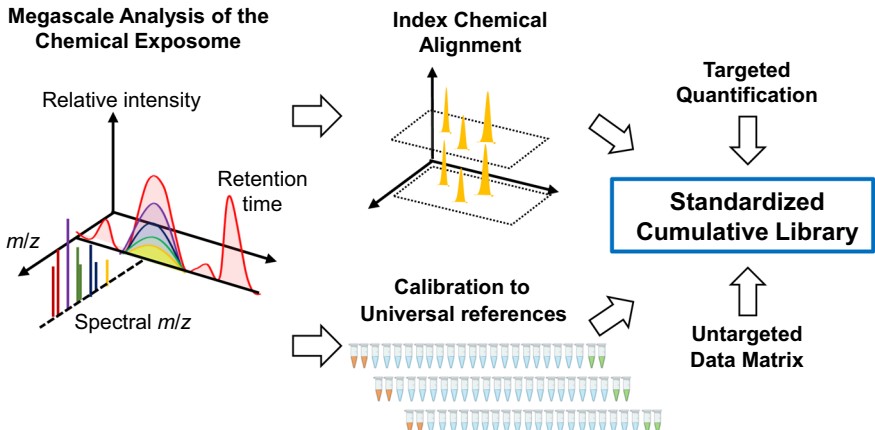

**Fig. 6 Application of XLE with GC–HRMS and reference standardization provides a framework for high-throughput exposome research.** The analytic workflow with single-step extraction and analysis along with pooled reference materials provides a simple, automatable method for measurement of low-abundance chemicals in biological materials. Left: The framework is anchored to accurate mass *m/z* signals that are clustered according to retention time and intensity correlations. Colors illustrate the presence of different *m/z* in spectra. Middle: These can be aligned relative to peaks of index chemicals (yellow) and quantified relative to standard reference materials (orange vials), thereby providing key criteria for interoperability and reproducibility. Right: Both targeted and untargeted chemical data obtained by this workflow are suitable for entry into cumulative data library to support exposome research.

determined by the exposome[31,32], yet detailed understanding is only available for a relatively small number of highly hazardous chemicals. Detection of less hazardous exposures requires large populations, and the cost for targeted assays of large numbers of chemicals generally precludes the study of large populations.

The presented XLE method can overcome these limitations by providing a standardized method to obtain quantitative information on known environmental chemicals while also providing information on thousands of unidentified chemicals. Based on a previously developed multidimensional framework using LC–HRMS to characterize the metabolome[27], XLE with GC–HRMS can be used to advance omics-scale analysis of the chemical exposome (Fig. 6). The simple XLE-processing procedure improves interoperability, and the multidimensional framework supports collection and communication of MS information for unidentified, as well as identified, chemicals for epidemiologic research. The information structure preserves key information about unidentified signals for subsequent chemical identification, i.e., accurate MS allows prediction of possible elemental compositions, isotopic internal standards are useful as index chemicals to facilitate retention-time alignment, and pooled reference materials provide quantitative reference and allow simultaneous identification and quantification of known environmental chemicals (Fig. 6). Storage of this information, along with other available

chemical and physical data, into standardized cumulative libraries, will provide a resource for exposome research.

Uncontrolled loss of chemicals is a major challenge in traditional sample preparation for analysis of environmental chemicals. Volatility, partitioning between solvents, and binding affinity to absorbents, vary substantially among different chemical classes. Multiple steps, including evaporation of solvent, adsorption, and reconstitution in traditional SPE and LLE methods, introduce variable loss of different chemicals. While this can be addressed by using stable isotope-labeled standards, this is impractical for omics scale analyses of environmental chemicals.

Several simple extraction methods have been developed to provide a more efficient alternative to multistep sample preparation, utilizing high-resolution mass spectrometry to expand the number of analytes. For example, GC–MS-based metabolomics using water/isopropanol/acetonitrile supports integration of targeted quantification of specific metabolites and untargeted discovery of novel compounds[33]. Simultaneous determination of multiple environmental chemicals based on liquid–liquid extraction has been developed yet usually requires a drying and reconstitution step to concentrate analytes[34,35]. While a dry-down step provides benefits in storage convenience and enrichment of low-abundance chemicals, it also introduces variability in chemical loss. In our experience testing liquid–liquid extraction

procedure, we found that chemicals have varied losses and recoveries in drying and reconstitution steps (shown in Fig. 1). Unless every single chemical has a stable isotopic counterpart in the procedure, this variability cannot be controlled. The drying step also extends the processing time, limiting throughput for large-scale studies.

The goal of XLE is to provide a streamlined workflow that supports quantification of known environmental chemicals while retaining information for untargeted identification of chemicals to enable population-level studies that require high-throughput analysis with limited sample volume. XLE minimizes uncontrolled variations by minimizing the number of sample-preparation steps. The results with three quantitative approaches, [13]C-labeled internal standards, external calibration curves, and reference standardization relative to NIST-1958, showed efficient recovery and quantification of PCBs, PBDEs, and chlorinated pesticides with XLE. Lower efficiencies are expected for dissimilar structures with greater polarity, a limitation which can be overcome by parallel use of LC–HRMS methods for exposome analysis[14]. Trade-offs for minimized sample preparation can include carryover from the sample matrices, which require stringent quality control and quality assurance with (1) real-time monitoring of brominated chemicals (e.g., PBDE-85 and PBB-153) that are sensitive to matrix interference; (2) multiple blank samples to support monitoring of peak baseline; and (3) frequent replacement and cleaning of the liner, column, filament, and ion source. In particular, the contamination of low-level biological samples by fortified samples needs to be assessed frequently by benchmarking against prior laboratory results and literature.

As previously shown for LC–HRMS data, reference standardization provides a practical approach for quantification relative to pooled reference material that is processed and analyzed concurrently with unknown samples. Reference standardization assumes a linear relationship between analyte concentration and instrument response, which was validated with authentic standards for the 378 chemicals reported here. Although signal response cannot be assumed to be consistent across matrices such as serum, plasma, urine, stool, and organ tissues, extrapolation based on well-calibrated human serum SRM provides a central and universal reference to support comparison and harmonization of a different dataset[36]. We demonstrated that this single-point calibration method performs comparably to quantification based on traditional six-point calibration or stable isotope-derived response factors for GC–HRMS in analyses of human plasma and lung (Fig. 3b, c, Supplementary Data 2 and 3). We also tested the levels of 4,4′-DDE, one of the most prevalent environmental chemicals, in serum and EDTA-treated plasma samples collected from the same 19 individuals and found high consistency of raw intensity in both matrices (Spearman's | $\rho$| = 0.70, $P < 0.001$, Supplementary Fig. 5). This result corroborated a previous comparison of plasma and serum for detection of perfluoroalkyl substances[37]. Calibration of more matrices as reference material and more chemicals other than halogenated chemicals reported in SRM-1957 and SRM-1958 will further improve accuracy and interoperability of reference standardization and provide expanded coverage in chemical quantification (Supplementary Data 6).

Using reference standardization, levels for environmental chemicals in the 80 CHDWB plasma samples were within the range of total population serum values reported by National Health and Nutrition Examination Survey (NHANES) in 2003–2004[22], e.g., 4,4′-DDE (0.971 [80 subject median] vs 1.29 [NHANES geometric mean] ng/g whole weight), HCB (0.032 vs 0.092 ng/g), trans-nonachlor (0.028 vs 0.089 ng/g), PCB-118 (0.021 vs 0.037 ng/g), and PCB-180 (0.066 vs 0.092 ng/g). The values for the CHDWB plasma are expected to be lower than

2003–2004 NHANES levels as the CHDWB participants were surveyed after 2006.

Human specimens other than blood samples are increasingly available from biorepositories and provide important opportunities for exposome research, as they may allow for assessments of exposure within target tissues of toxicity[38,39]. Historically, white adipose tissues were sampled as a storage and effector site for persistent environmental pollutants[40]. As environmental factors are now recognized to contribute to the origins and expressions of many human diseases, more extensive analysis of clinically annotated tissue samples is needed. The present results showing quantification of multiclass chemicals demonstrate that XLE and reference standardization provide a generalizable approach for human specimens. Compared with human plasma, the more volatile PCBs and HCB were more predominant in the lung, indicating the potential effects of respiratory exposure to organ chemical profiles (Supplementary Data 8). On the other hand, subjects with severe obesity showed lower plasma levels of common persistent pollutants than subjects with normal BMI ($P < 0.01$ for HCB, PCB-138, -153, -170, and -180), indicating a negative effect of body fat on circulating levels of lipophilic chemicals[41]. Stool had a relatively low environmental chemical content, but is important as a route for elimination of lipophilic chemicals from fat reservoirs[25] and source for information on recent exposures from diet[42]. The results obtained reflect inter-organ interactions in chemical uptake, distribution, and clearance, and stress the value of XLE with GC–HRMS for analysis of diverse human tissue types. GC–MS has also been used in urinary analysis of biomarkers of exposure to different classes of chemicals[43]. In principle, XLE-based GC–HRMS can be extended to quantitative measurements of trace levels of the more hydrophobic and volatile precursors in large-scale cohort studies, such as precursors for hydroxylated PCBs[44], hydroxylated PAHs[45], polar pesticides, and metabolites[46]. This would complement assessments of more polar xenobiotic metabolites that have been developed for LC–MS analyses of urine.

XLE with GC–HRMS provides an important step forward for human exposome research by providing a method to maximize capture of information on unidentified chemicals in human samples. Use of such information is facilitated by available annotation and library-search software and algorithms, such as RAMclustR and MS-Finder. These tools cluster spectral ions and predict potential structure information on unidentified spectral features. Even without identification, statistical tests can be applied to detect mass spectral features associated with health outcomes. Unlike LC–MS analysis where adduct forms of intact molecular ions are routinely measured, GC–MS uses high-energy electron ionization (EI) and the associated spectra often lack an intact molecular ion and require deconvolution for chemical identification. Although METLIN[47,48] is one of the most comprehensive databases that covers a wide range of small molecules and MS/MS spectra collected by ESI, information on high-resolution GC-MS of environmental chemicals is more limited. Other GC–MS databases contain either low-resolution spectra[49] or mainly endogenous metabolites[50]. The lack of high-resolution databases for environmental chemicals is a major challenge and necessitates development of more high-resolution environmental contaminant libraries. A rigorous interoperable data-reporting structure will enable research-community efforts to advance chemical identification.

In conclusion, XLE with GC–HRMS addresses a critical need for methods to deliver omics-scale biomonitoring data for exposome epidemiology in an automatable, high-throughput and affordable manner. The method provides measures for both known and unidentified MS features to test for associations with disease. For known chemicals, an automated workflow integrates computational methods for data extraction, preprocessing and spectral annotation; with

reference standardization, the method supports quantification of environmental chemicals. Tests with plasma, lung, thyroid, and stool samples showed that the method is suitable for multiple sample types. The simplicity of the method facilitates harmonization of exposome analyses, enabling development of cumulative human exposome databases to include information on tens of thousands of chemical exposures in tens of thousands of individuals.

## Methods

**Standards and reference materials.** As an initial screening, we have purchased and examined spectra and chromatographic information for over 900 authentic chemical standards (1–20 ng/mL) in the form of a single chemical or a mixture of ≤40 chemicals, from Cambridge Isotopes (Tewksbury, MA) and AccuStandard (New Haven, CT). A total of 556 chemicals showed high detection sensitivity and linearity ($|\rho| > 0.98$ over 0–20 ng/mL); 378 of these analyzed under dilution conditions in isooctane (0.05–2 ng/ mL) relevant for human analyses were entered into a data library (Supplementary Data 4). In addition, two $^{13}C$-labeled chemicals [$^{13}C_{12}$]PCB-28 and [$^{13}C_{12}$]PBB-153 were used as volumetric internal standards added to the final extract, and nine $^{13}C$-labeled chemicals (99% isotope enrichment for each) were spiked as recovery standards to estimate chemical recovery efficiency by XLE: [$^{13}C_{12}$]PCB-101, [$^{13}C_{12}$] PCB-153, [$^{13}C_{12}$]PCB-180, [$^{13}C_{12}$]PBDE-47, [$^{13}C_{12}$]PBDE-99, [$^{13}C_6$]anthracene, [$^{13}C_{10}$]mirex, [$^{13}C_6$]cis-permethrin, and [$^{13}C_{12}$]4,4'-DDE.

National Institute of Standards & Technology [NIST] Standard Reference Materials (SRM) 1958 was analyzed with ≥3 aliquoted replicates per experiment to validate and test reproducibility of detection and quantification. Two sets of SRM, NIST SRM-1958 (human serum fortified with environmental chemicals) and NIST SRM-1957 (nonfortified human serum) were run in every batch of 20 samples to support quality control and quantification using reference standardization, a protocol that was previously validated in high-resolution mass spectrometry data for LC methods[17]. In this protocol, individual chemical concentrations in unknown samples are estimated by comparison with a concurrently analyzed, pooled reference sample with known chemical concentrations.

**Human materials.** Ethylenediaminetetraacetic acid (EDTA)-treated plasma samples were collected following standard operating procedures from 80 individuals without known disease and were randomly selected from archival samples obtained from the Center for Health Discovery and Well Being (CHDWB) cohort of approximately 750 individuals. The original study was conducted under Emory Investigational Review Board (IRB approval No. 00007243) and included both genders and individuals self-identifying as white, black, Hispanic, and Asian[51–53]. The inclusion and exclusion criteria of CHDWB were structured to identify a cohort of adults with few known acute or uncontrollable chronic conditions in order to represent a healthy population relative to the characteristics of the general US population. Individuals were excluded if they were functionally impaired by poorly controlled chronic disease or acute illness, but included if taking medications for common ailments. Plasma samples used for the present study included samples collected at the time of recruitment (baseline), and from visits six and 12 months later for follow-up. Participants were provided personalized counseling regarding promotion of a healthier lifestyle, yet all health behavior changes (if any) are entirely at the discretion of the individual. The informed consent contains wording to allow investigators to perform measurements and access to the participant's health-related data for research purposes at the discretion of the investigators or center staff and thus the present study. The primary outcomes of this study have been published previously[51–53].

Whole human lungs were from eleven individuals, four were deidentified end-stage diseased lungs acquired from the Emory Transplant Center (IRB approval No. 00006248), one was a deidentified post-transplant lung from Cystic Fibrosis Biospecimen Registry at Emory University (IRB approval No. 00095116), and six nondiseased deidentified postmortem lungs were obtained through the International Institute for the Advancement of Medicine (IIAM, Edison, NJ) or Novabiosis (Morrisville, NC). For post-transplant human donor lungs, de-identified tissue was collected by the Emory Transplant Center under Institutional Review Board approved tissue-acquisition protocols (IRB approval No. 00006248) and Cystic Fibrosis Biospecimen Registry at Emory University (IRB approval No. 00095116) with patient informed consent, which gives permission to use discarded tissues from organ-transplant patients for assay development and thus the present study. Whole human thyroids from five individuals were postmortem tissues that were acquired from National Disease Research Interchange (NDRI, Philadelphia, PA). Collection and research using postmortem organs (i.e., the six lungs and all thyroids in this study) in the United States do not represent human research activity since the samples are not from living individuals, and thus waiver of patient consent has been approved by Emory Institutional Review Board in accordance with 45 CFR Part 46.102 (f) and 21 CFR 50.3 (g). Human stool samples were obtained from PROGRESS, a biobank for cholestatic liver disease (IRB 670-02; Mayo Clinic, Rochester, MN) with informed consent for future use of collected patient samples in biomedical research. The use of stool samples from patients with primary sclerosing cholangitis (PSC) in the current study was approved (IRB 14-008752; Mayo Clinic, Rochester, MN) and additional informed consent

was waived in accordance with 45 CFR 46.116, as the study is of minimal risk to subjects.

**XLE sample extraction.** For plasma samples, 50 μL of formic acid (Emprove® Essential DAC, Sigma-Aldrich) was added to 200 μL of plasma and immediately followed by addition of 200 μL of hexane–ethyl acetate (2:1 v/v, ≥99% pure, Sigma-Aldrich) containing the internal standards (final concentration: 1 ng/mL). The sample mixture was shaken vigorously on ice using multitube vortexer (VWR VX-2500) for 1 h and centrifuged at 1000 $g$, 4 °C for 10 min. The sample mixture was chilled during the entire extraction procedure. The organic supernatant was transferred to a new tube with 25 mg of MgSO₄ (≥99.99% pure, Sigma-Aldrich), or dSPE (150 mg of MgSO₄, 50 mg of PSA, 50 mg of C18-EC, Restek Catalog #26125) for testing of QuEChERS based procedure, and vortexed vigorously to remove water. After 10 min of centrifugation at 1000 $g$, 80 μL of the final supernatant was spiked with volumetric internal standards (final concentration: 1 ng/mL) for analysis. Tissue samples were processed after homogenization in 250–300 μL of water and extracted with 50 μL of formic acid with hexane–ethyl acetate mixture in a ratio of 4:1 (lung) or 5:1 (other samples) in proportion to sample weight: 100 mg of homogenized lung with 400 μL of hexane–ethyl acetate mixture, 40 mg of homogenized thyroid with 200 μL of hexane–ethyl acetate mixture, and 100 mg of stool samples with 500 μL of hexane–ethyl acetate mixture, and then processed as plasma samples. The variation of 4:1 from 5:1 in lung extraction was arbitrary in consideration of the lower density of lung as an organ. Samples were prepared in batches containing 20 samples along with two SRM samples, two isooctane blanks, two solvent blanks containing internal standards, and two saline samples that went through solvent extraction, as part of quality-control measures.

**Instrumental analysis.** Samples were analyzed with three injections for plasma and two injections for other specimen using GC–HRMS with a Thermo Scientific Q Exactive GC hybrid quadrupole mass spectrometer with 2 μL per injection. A capillary DB-5MS column (15-m × 0.25-mm × 0.25-μm film thickness) was used with the following temperature program: hold 75 °C for 1 min, 25 °C/min to 180 °C, 6 °C/min to 250 °C, 20 °C/min to 350 °C, and hold for 5 min. The flow rate of the helium carrier gas was 1 mL/min. Ion source and transfer-line temperatures were 250 °C and 280 °C, respectively. Data were collected from 3 to 24.37 min with positive electron-ionization (EI) mode (+70 eV), scanning from $m/z$ 85.0000 to 850.0000 with a resolution of 60,000.

**Quality control and assurance.** To avoid negative impact from minimized sample clean-up step, we follow the following standard operating procedures: (1) peak intensities of 12 selected chemicals are monitored real time in every SRM-1958, including 4,4′-DDE, 4,4′-DDT, HCB, α-, β-, γ-BHC, PCB-138, PCB-153, PCB-180, PBDE-47, PBDE-85, PBDE-153, and PBB-153, which are found to be sensitive to matrix interference; (2) contamination and carryover were checked in isooctane washes, solvent blanks, and method blanks that are run at the beginning of each batch by monitoring of peak baseline; and (3) routine maintenance includes changing the liner for every 60 injections of study samples (i.e., one batch), changing the filaments, the column and cleaning the ion source every 480 injections of study samples (i.e., eight batches) or every new study.

**Data extraction and preprocessing.** Raw data were examined by checking the signal-to-noise ratio, peak shape, and spectral information for surrogate and internal standards using a 5-ppm $m/z$ tolerance and a 30-s retention-time window in XCalibur Qualbrowser software. TraceFinder software version 4.1 (Thermo-Fisher Scientific) was tested with mixtures of standards and found challenging to simultaneously detect >250 chemicals. Thus, data extraction was performed by XCMS[54] to generate about 40,000 chemical features identified by spectral $m/z$ and retention time; extraction with apLCMS[55] generated more than 200,000 $m/z$ features that were considered too many for current needs. Data were prefiltered to retain around 25,000 features that had average peak intensities for nonblank samples that were 10-fold greater than saline-method blanks.

For targeted quantitation, we used the library of 378 chemical standards that included spectral information (the five most abundant $m/z$) and retention time (Supplementary Data 4). Features were selected with tolerance of ±5 ppm $m/z$ and 30 s of retention time by GetVenn function in xMSAnalyzer[56], and further clustered by RAMclustR[28] based on feature similarity in retention time and correlation across samples. Features were matched to chemical spectra for identification, and intensities of the most abundant $m/z$ fragments were used for quantification. Alternatively, as an untargeted approach, features were clustered with RAMclustR without matching to target chemicals to support biostatistics and bioinformatics analysis before chemical annotation and identification. Spectral clusters generated from RAMclustR were searched against chemical spectral libraries using MS-Finder (ver 3.42)[29,30] and XCalibur v 4.2.1. All codes of data extraction and analysis are presented in Supplementary Data 11.

**Metabolite quantification in NIST reference serum using external standard curves.** Recoveries of each recovery standard were determined after normalizing to [$^{13}C_{12}$]PCB-28 added as a volumetric internal standard in SRM-1957 samples ($n = 48$). For each authentic environmental chemical, a serial dilution of 0, 0.05,

0.1, 0.25, 0.5, and 1 and 2 ng/mL in hexane–ethyl acetate mixture (2:1, v/v, $n = 3$ each) was run and analyzed through data extraction and preprocessing together with SRM-1958 ($n = 9$) and SRM-1957 ($n = 9$). Instrument detection limits and linearity of response were calculated from the diluted standards[57] using the most abundant spectral $m/z$ (Supplementary Data 4). Absolute quantification of chemicals in SRM-1958 was determined from external calibration curves using the most abundant $m/z$ fragment in full-scan mode.

**Reference standardization**. Reference standardization is a simple quantification protocol in which individual chemical concentrations in unknown samples are estimated by single-point calibration to a concurrently analyzed, pooled reference sample with known chemical concentrations. This method has been previously validated for LC–HRMS data[17,36]. To validate the use in GC–HRMS, a randomly selected set of human plasma ($n = 20$) was quantified by two methods: (1) Reference standardization using NIST SRM-1958 and NIST SRM-1957 that were run in parallel to unknown samples; for chemicals that were detected in both SRM-1957 and SRM-1958, the certified mass fractions from both references were used by plotting an unweighted-linear regression line (X axis = chemical concentration in references, Y axis = chemical peak area in references); for chemicals that were only detected in SRM-1958 and chemicals with a positive x intercept of the linear-regression line, the certified mass values from SRM-1958 were used with the following equation: concentration$_{sample}$ = concentration$_{reference}$ × (intensity$_{sample}$/intensity$_{reference}$). (2) External calibration curves constructed by known concentrations of standards (0, 0.05, 0.1, 0.25, 0.5, and 1 and 2 ng/mL in hexane–ethyl acetate mixture) and adjusted for recoveries measured by [$^{13}$C]-labeled internal standards. To compare the matrix effect on quantification, a separate set of EDTA-treated plasma and serum collected from the same individuals ($n = 19$) was processed and raw intensities of the primary ion of 4,4′-DDE levels were measured using XLE–GC–MS workflow. Reference standardization was further validated in analyses of human lungs ($n = 11$). Quantification of six chemicals that were detected frequently in the lung and had a [$^{13}$C] labeled isotopic counterpart spiked were performed using the RF based EPA method[24]. RF were determined by certified mass fraction of chemicals and measured intensity of [$^{13}$C] labeled isotopes in SRM-1958 ($n = 2$) that were processed in parallel to the lung samples.

After reference standardization based on SRM-1957 and SRM-1958 was validated, quantification results of human plasma, tissue, and stool samples were obtained using this approach as described above[17,36]. In each batch, SRM-1957 and SRM-1958 were extracted and analyzed in parallel to study samples to support batchwise quantification.

**Statistics and reproducibility**. Data are presented as the mean and standard error of the mean. Sample size ($n$) represents the number of distinct samples and measurement for each sample was mean values from repeated injections. Welch's $t$ test was used to determine significant differences between two groups, and one-way ANOVA was used among multiple groups. Mann–Whitney U test was used as a nonparametric test when normality assumption failed by Shapiro–Wilk test ($P < 0.05$). Spearman's correlation analysis was used instead of Pearson's correlation when normality assumption failed. All were performed with SigmaPlot 14.0 (Systat Software, Inc). Unsupervised two-way hierarchical clustering was performed and visualized in heatmap using MetaboAnalyst v4.0[58] using Pearson's clustering algorithm on log-transformed raw intensity. All other bioinformatics analyses were performed in R Studio version 1.1.447 (RStudio, Inc). Graphs were generated by SigmaPlot 14.0 or R (v3.6.1, code provided in Source Data). Illustrations in Figs. 3a, 5a, and 6 were created with BioRender.com. The significance level was $p < 0.05$ for all tests. XLE procedure optimization experiments were repeated at least three times at sets of $n = 3$–4 and representative results were presented. Recovery of internal standards and quantification of chemicals in SRM were repeated two times and all results were included and combined for a total of $n = 9$–16. Reference-standardization validation in human plasma was repeated with a different subset ($n = 20$) of samples, and quantification of chemicals in lung was repeated twice in a subset of two lung samples. All replications generated similar results.

**Reporting summary**. Further information on research design is available in the Nature Research Reporting Summary linked to this article.

## Data availability
The raw data generated in this study have been deposited in the National Institutes of Health (NIH) Common Fund's National Metabolomics Data Repository (NMDR), the Metabolomics Workbench (https://www.metabolomicsworkbench.org) under assigned Project ID PR001136, accessible at https://doi.org/10.21228/M8VQ4D[59]. The Metabolomics Workbench is supported by NIH grant U2C-DK119886. Source data are provided with this paper.

## Code availability
All open-source software used in this study was R-based packages that have been previously published and can be downloaded at https://bioconductor.org/packages/release/bioc/html/xcms.html (XCMS); https://sourceforge.net/projects/xmsanalyzer/

(xMSanalyzer); https://github.com/cbroeckl/RAMClustR (RAMClustR). Parameter settings of the software in the present study are provided in Supplementary Data 11.

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

## Acknowledgements

This study was supported by the NIEHS, U2C ES030163 (DPJ), R24 ES029490 (DPJ, XH), U2C ES030859 (DIW) and P30 ES019776 (CJM), NIDDK RC2 DK118619 (KNL), NHLBI R01 HL086773 (DPJ), US Department of Defense W81XWH2010103 (DPJ), and the Chris M. Carlos and Catharine Nicole Jockisch Carlos Endowment Fund in Primary Sclerosing Cholangitis (PSC) (KNL).

## Author contributions

X.H., D.I.W., C.J.M., Y.M.G., K.D.P., G.W.M., K.N.L. and D.P.J. discussed and designed the study. X.H., D.I.W. and Y.L.L. tested the analytical method and performed experiments. M.R.S., M.L.O., B.D.J., M.K., G.S.M., D.C.N. and K.N.L. collected and provided human material. C.Y.M. and K.U. tested and optimized the parameters of data extraction. X.H. and D.P.J. analyzed and interpreted data. X.H., D.I.W., C.J.M., K.D.P., G.W.M., K.N.L. and D.P.J. prepared and edited the paper text. All authors reviewed the paper.

## Competing interests

The authors declare no competing interests.
