## [Peer Review File · Nature Communications]

REVIEWER COMMENTS

Reviewer #1 (Remarks to the Author):

It is a strength that the quantification of environmental chemicals was shown in diverse human samples, including plasma, lung, thyroid and stool. Urine would be good as well as it is most often used for exposure assessment in cohort studies and since it is especially challenging to ensure consistent analytical performance (varying matrix effects and interferences depending on the dilution of the individual urine sample).

· Using the NIST reference material (SRM-1957 (freeze-dried non-fortified human serum) and SRM-1958 (freeze-dried fortified human serum)) for benchmarking performance is of course great ('Overall, XLE supported measurement of 71 out of the 73 chemicals that are in the ng/kg range in SRM-1958'). However, I am slightly concerned that the evaluation was based on a single reference material and included only halogenated compounds (which have kind of a similar chemistry and some advantages with regards to isotope pattern that can be used for better identification).

· 'We analyzed 20 human plasma samples and performed reference standardization of 17 chemicals based on SRM-1957 and SRM-1958 that were processed in parallel to the plasma samples.' In our experience and according to literature it is not feasible to compare plasma and serum, especially not when it has been freeze dried. In addition, the information on the kind of plasma (citrate, heparin, EDTA) is missing but highly relevant in HRMS experiments

· The 'Application of the XLE workflow for analysis of environmental chemicals in human samples' is interesting but I do not fully understand the following sentence: 'For targeted environmental chemical analysis, we selected 378 chemicals from an in-house database for which dilution conditions (0.05 to 2 ng/mL) were relevant for general-population analyses (Supplemental Table S3).'; kindly consider re-phrasing

· XLE sample extraction: 'For plasma samples, 50 µL formic acid (Emprove® Essential DAC, Sigma-Aldrich) was added to 200 µL plasma and immediately followed by addition of 200 µL hexane – ethyl acetate (2:1 v/v, ≥99% pure, Sigma-Aldrich) containing the internal standards (final concentration: 1 ng/mL). The mixture was shaken vigorously on ice using multi-tube vortexer (VWR VX-2500) for 1 h and centrifuged at 1000 g, 4 °C for 10 min.' I am wondering if this protocol is really suited for exposome analysis as it could result in losses of acid-labile and temperature sensitive analytes. Or was the mixture chilled during the 1 h extraction step? Please discuss.

· The quality control should be described and discussed in more detail. How often does the liner need to be changed and how does this impact very long sequences (as this is the aim of any exposome methodology)?

· A common issue in trace analysis is the contamination of biological samples intended as 'blank samples' in fortification experiments. How do the authors address this issue (especially when aiming for quantitative data) and how is there general advice how to approach with ever increasing sensitivity of mass specs (also in LC-HRMS)?

· 'Stool samples (100 mg) were homogenized and extracted directly in 50 μ L formic acid and 200 μ L solvent and then processed as plasma samples.' How many human stool samples were analysed and why was the volume of extraction solvent not corrected by the dry weight? For my feeling 100 mg is quite a high amount for 250 μ L solvent, we typically apply a ratio of 1:10 – 1:100 for stool, otherwise the extraction might be impacted and the mass spec pretty dirty following a long sequence (with decreasing signals)

· All raw and processed data should be deposited in public repositories (FAIR principles) including the NIST material measurements and the reference standards

Reviewer #2 (Remarks to the Author):

The paper "A scalable workflow for the human exposome" by Hu et al. is a methodology-rich paper that looks to produce a simple extraction and GC-HRMS workflow. The authors were able to produce reproducible GC-HRMS data, using multiple (2) users and were able to extract metabolites from a number of starting materials. The conclusions of this paper are supported by the work undertaken and seem sound. Although, it would be good to see how the results produced compare to those in the METLIN Exposome Database.

The GC-HRMS method is logical. However, the authors propose using the full scan method rather than SIM method for making sure all metabolites are identified. I wonder if using a simultaneous Scan/SIM acquisition method would help with metabolite identification of the data.

The authors, in a bid to produce single-step liquid extraction technique, did not do a final solid phase extraction step nor liquid-liquid extraction. I wonder if this had any adverse effects on the acquisition of data from the GC-HRMS. For instance, there would have, inevitably, been underivatized fatty acids carried over from the sample extraction. These can cause problems and lead to faster degradation of the column. The [13C12]p,p'-dichlorodiphenyldichloroethylene (p,p'-DDE) reference sample only had a recovery of less than 70 %. This suggests optimisation is needed as you would expect recoveries of 70-130 % for well optimised protocols. Is this due to a lack of sample clean up? Lack of optimisation of solvent/column used? Although, I do applaud the authors rational for carrying on with the method.

Interestingly, the authors suggest that the solvents and reagents used for QuEChERS caused contamination (which I don't see from Figure 2A). As I understand it, the QuEChERS method (and its derivatives) is a standard protocol and I wonder what contaminations are being seen in this experiment? And how it may affect other work?

The methods are detailed enough to be reproducible. For the sake of clarity and reproducibility, it would be better for the term "solvent" in the "XLE Sample Extraction" sub-section of the "Material and Methods" to be replaced with more appropriate wording to indicate the hexane-ethyl acetate mixture was used.

Reviewer #3 (Remarks to the Author):

In this manuscript, Hu et al describe an optimized extraction method to quantify small molecules in plasma and tissue samples by GC-MS, with a focus on exposomic applications. Untargeted quantification of the exposome, particularly in tissue samples, is of great relevance. However, I have several concerns.

1. Novelty: Although their method is easy and robust, there are many simple extraction methods for GC-MS. Even though these methods are not necessarily applied specifically to the study of the exposome, there is no reason to assume that they would be unsuitable. Specific examples that are easy and do not include a drying step include PMID: 27038389, PMID: 29204936, PMID 29256082. Likewise, a dry-down step is often included by researchers as a convenience for storage between extraction and MS run, or for cross-sample normalization. Thus, even though many published methods include such a dry-down method, there is no reason to assume that it would not be disposable in prior publications (e.g. PMID: 24631817).

2. Reproducibility:

2a. Methods are insufficient to enable full reproducibility with regards to data processing. All XCMS parameters should be provided. I also question whether these parameters were appropriately optimized – 40,000 features seems excessive for serum (even 25,000 after blank removal is still very high).

2b. any necessary RAMclustR parameters should be provided

2c. Data availability: “Raw data of this study is in the process of being uploaded to Metabolomics Workbench” is insufficient. Please provide accession number.

2d. Please consider providing R code for all analyses and figure generation. While not a requirement, this would facilitate reproducibility.

2e. “Starting initially with a QuEChERS procedure (19, 20), we systematically varied solvent composition, volume and extraction time to obtain a simplified procedure with a minimal number of steps and possibilities for contamination and variability in recovery”. Presenting that data would facilitate reproducibility, and would also enable other researchers to build on these optimization steps. As currently presented, the rationale for the selection of the optimized XLE method is unknown.

3. Approach:

3a. Extraction ratios are inconsistent across sample types. The authors should provide a rationale for why these different values were selected rather than a standard ratio based on dry tissue weight.

3b. Using these different extraction ratios likely leads to great variability in terms of chemical concentration and matrix effects across sample types. Thus, comparison across sample types is inappropriate, as is performed in Figure 4C. This issue is compounded by the authors’ normalization of tissue data to serum standards: signal intensity and/or standard curve response cannot be assumed to be consistent across matrices. Reference standardization is only appropriate for samples vs references of the same type.

3c. For best comparability, standards should be prepared in the same solvent as sample extracts, rather than in isoctane.

4. Data interpretation and statistics:

4a. y axis in fig 2E is misleading. The larger values for p,p'-DDE make it appear like the measured vs certified values are similar for all the other chemicals, when in fact several are only at ~75% for PCBs.

4b. Likewise, the authors bias the interpretation of this data by only selecting the most successful PCBs from Table S1 to be plotted. Indeed, many are highly divergent (e.g. PCB 149)

4c. Although the authors (rightly) criticize the excessive focus of exposomics studies on a small list of known chemicals, they too focus predominantly on a restricted list of 14-49 compounds in participant samples and only briefly address additional compounds. Section “Analysis of unidentified

environmental chemicals” would benefit from additional annotation efforts, and from analysis of collected tissue data.

4d. Fig 3C: did the authors consider normality? Pearson correlation may not be the most appropriate here.

Minor concerns:

1. Abstract: 200 μ L plasma and 100 mg tissue samples are not small amounts.
2. Figure 1A is uninformative.
3. Figure 4B data: It is relatively unsurprising that chemicals from the same class would be correlated to each other since they likely derive from similar environmental exposure events.
4. Typo in table S4 title “intensity”
5. Spearman correlation is used in results but not mentioned in Methods.
6. Line and page numbers would be appreciated in the future.

RESPONSE TO REVIEWER COMMENTS

Reviewer #1 (Remarks to the Author):

It is a strength that the quantification of environmental chemicals was shown in diverse human samples, including plasma, lung, thyroid and stool. Urine would be good as well as it is most often used for exposure assessment in cohort studies and since it is especially challenging to ensure consistent analytical performance (varying matrix effects and interferences depending on the dilution of the individual urine sample).

We have included a paragraph (page 18, lines 373-378) addressing use of XLE for urine analysis of hydrophobic and volatile environmental chemicals to complement the measurement of more hydrophilic xenobiotic metabolites often measured in urine using targeted analysis by LC-MS.

· Using the NIST reference material (SRM-1957 (freeze-dried non-fortified human serum) and SRM-1958 (freeze-dried fortified human serum)) for benchmarking performance is of course great ('Overall, XLE supported measurement of 71 out of the 73 chemicals that are in the ng/kg range in SRM-1958'). However, I am slightly concerned that the evaluation was based on a single reference material and included only halogenated compounds (which have kind of a similar chemistry and some advantages with regards to isotope pattern that can be used for better identification).

Reference material on environmental chemical analysis is limited. We have now included results (page 7, lines 149-152) of an additional reference material: SRM-1957 (non-fortified human serum) which show XLE supported detection and reproducible measurement of 29 out of 32 chemicals with certified or estimated reference values in the ng/kg range in SRM-1957.

As now stated in the text, only halogenated compounds have reference values reported in SRM-1957 and SRM-1958. We have expanded our library to contain a broad range of non-halogenated compounds, now included in a new Supplemental Table S6. These include PAHs, organophosphate pesticides, by-products or intermediates in commercial manufacturing. We also reported detection and quantitative intensity of these chemicals in human plasma in Supplemental Table S7.

· 'We analyzed 20 human plasma samples and performed reference standardization of 17 chemicals based on SRM-1957 and SRM-1958 that were processed in parallel to the plasma samples.' In our experience and according to literature it is not feasible to compare plasma and serum, especially not when it has been freeze dried. In addition, the information on the kind of plasma (citrate, heparin, EDTA) is missing but highly relevant in HRMS experiments.

We have now discussed these important issues in the text (page 17, lines 340-348). The plasma analyzed in this manuscript were all EDTA treated plasma, collected following standard operating procedure (page 21, lines 428-429). As now clarified, use of common reference materials facilitates harmonization of analyses between laboratories but is subject to matrix effects.

· The 'Application of the XLE workflow for analysis of environmental chemicals in human samples' is interesting but I do not fully understand the following sentence: 'For targeted environmental chemical analysis, we selected 378 chemicals from an in-house database for which dilution conditions (0.05 to 2 ng/mL) were relevant for general-population analyses (Supplemental Table S3).'; kindly consider re-phrasing

We have revised this sentence as requested, page 9, lines 179-181.

· XLE sample extraction: 'For plasma samples, 50 µL formic acid (Emprove® Essential DAC, Sigma-Aldrich) was added to 200 µL plasma and immediately followed by addition of 200 µL hexane – ethyl acetate (2:1 v/v, ≥99% pure, Sigma-Aldrich) containing the internal standards (final concentration: 1 ng/mL). The mixture was shaken vigorously on ice using multi-tube vortexer (VWR VX-2500) for 1 h and centrifuged at 1000 g, 4 °C for 10 min.' I am wondering if this protocol is really suited for exposome analysis as it could result in losses of acid-labile and temperature sensitive analytes. Or was the mixture chilled during the 1 h extraction step? Please discuss.

Additional description is now provided (page 22, lines 462-464).

· The quality control should be described and discussed in more detail. How often does the liner needs to be changed and how does this impact very long sequences (as this is the aim of any exposome methodology)?

We have now added details of our quality control procedures (page 23, lines 488-496).

· A common issue in trace analysis is the contamination of biological samples intended as 'blank samples' in fortification experiments. How do the authors address this issue (especially when aiming for quantitative data) and how is there general advice how to approach with ever increasing sensitivity of mass specs (also in LC-HRMS)?

We have clarified our use of blanks and reference materials to address these issues (page 23, lines 489-494). We have also added additional advice concerning use of prior laboratory and literature for benchmarking results (page 16, lines 322-329).

· 'Stool samples (100 mg) were homogenized and extracted directly in 50 µL formic acid and 200 µL solvent and then processed as plasma samples. 'How many human stool samples were analysed and why was the volume of extraction solvent not corrected by the dry weight? For my feeling 100 mg is quite a high amount for 250 µL solvent, we

typically apply a ratio of 1:10 – 1:100 for stool, otherwise the extraction might be impacted and the mass spec pretty dirty following a long sequence (with decreasing signals)

We appreciate with this concern and apologize for the error. Initially we used 100 mg stool samples with 50 μ L formic acid and 200 μ L solvent. It turned out the mixture was too viscous to separate into distinct layers, and additional 300 μ L solvent was immediately added to the extraction. Therefore, we actually used a ratio of 1:5 (stool to solvent) with addition of 50 μ L formic acid. No decreasing of signals were found after our small sample of testing. However, given the sufficient level of peak intensities, it is applicable to use a ratio of 1:10. We have corrected and updated the information in the revision (page 23, lines 474).

· All raw and processed data should be deposited in public repositories (FAIR principles) including the NIST material measurements and the reference standards

All raw and processed data have been deposited into the Metabolomics Workbench. Project ID and DOI is now provided in main text (page 27, lines 581-582).

Reviewer #2 (Remarks to the Author):

The paper “A scalable workflow for the human exposome” by Hu et al. is a methodology-rich paper that looks to produce a simple extraction and GC-HRMS workflow. The authors were able to produce reproducible GC-HRMS data, using multiple (2) users and were able to extract metabolites from a number of starting materials. The conclusions of this paper are supported by the work undertaken and seem sound. Although, it would be good to see how the results produced compare to those in the METLIN Exposome Database.

We appreciate this comment and have added discussion (page 19, lines 385-391). Unlike LC-MS analysis where adduct forms of intact molecular ions are routinely measured, GC-MS uses high-energy electron ionization (EI) and the associated spectra often lack an intact molecular ion and require deconvolution for chemical identification. Although METLIN is one of the most comprehensive databases that covers a wide range of small molecules and MS/MS spectra collected by ESI, information on high-resolution GC-MS of environmental chemicals is more limited. Other GC-MS databases contains either low-resolution spectra or mainly endogenous metabolites. The lack of high-resolution database on environmental chemicals is a major challenge and necessitated development of an in-house library for developing this exposome research workflow.

The GC-HRMS method is logical. However, the authors propose using the full scan

method rather than SIM method for making sure all metabolites are identified. I wonder if using a simultaneous Scan/SIM acquisition method would help with metabolite identification of the data.

As now discussed (page 4, lines 72-75) the high-field Orbitrap detector used in this study has sufficient sensitivity and selectivity in full MS mode and provides the greatest benefit, in terms of ions and compounds monitored, because the time of “scan” (or data collection) spent in full scan mode is reduced by half (time for switching to SIM) when using synchronous SIM/scan. Thus, synchronous SIM/scan reduces sensitivity on chemicals that are not monitored by SIM simultaneously.

The authors, in a bid to produce single-step liquid extraction technique, did not do a final solid phase extraction step nor liquid-liquid extraction. I wonder if this had any adverse effects on the acquisition of data from the GC-HRMS. For instance, there would have, inevitably, been underivatized fatty acids carried over from the sample extraction. These can cause problems and lead to faster degradation of the column. The [13C12]p,p'-dichlorodiphenyldichloroethylene (p,p'-DDE) reference sample only had a recovery of less than 70 %. This suggests optimisation is needed as you would expect recoveries of 70-130 % for well optimised protocols. Is this due to a lack of sample clean up? Lack of optimisation of solvent/column used? Although, I do applaud the authors rational for carrying on with the method.

This is an important issue and, as now discussed in more detail (page 16, lines 322-329), especially negative impacts due to carry-over from the sample matrices. Tradeoffs for the single-step extraction to minimize the uncontrolled variability in chemical recovery require stringent quality control and quality assurance with 1) real-time monitoring of brominated chemicals that are sensitive to matrix interference; 2) multiple blank samples to support monitoring of peak baseline and 3) frequent replacement and cleaning of liner, column, filament and ion source.

Interestingly, the authors suggest that the solvents and reagents used for QuEChERS caused contamination (which I don't see from Figure 2A). As I understand it, the QuEChERS method (and its derivatives) is a standard protocol and I wonder what contaminations are being seen in this experiment? And how it may affect other work?

As now clarified (page 6, lines 111-116 and Supplemental Figure S1), we did not see substantial difference between the two cleaning methods. Compared to QuEChERS – dSPE, high purity MgSO₄ showed similar reproducibility. Because it may provide benefit for certain targeted chemicals, such as low-intensity PBB-153, and avoided variable contamination by uncharacterized chemicals, we decided to use MgSO₄ to remove interferences.

The methods are detailed enough to be reproducible. For the sake of clarity and reproducibility, it would be better for the term “solvent” in the “XLE Sample Extraction” sub-section of the “Material and Methods” to be replaced with more appropriate wording to indicate the hexane-ethyl acetate mixture was used.

We have replaced the wording as suggested (page 22, lines 470-476).

Reviewer #3 (Remarks to the Author):

In this manuscript, Hu et al describe an optimized extraction method to quantify small molecules in plasma and tissue samples by GC-MS, with a focus on exposomic applications. Untargeted quantification of the exposome, particularly in tissue samples, is of great relevance. However, I have several concerns.

1. Novelty: Although their method is easy and robust, there are many simple extraction methods for GC-MS. Even though these methods are not necessarily applied specifically to the study of the exposome, there is no reason to assume that they would be unsuitable. Specific examples that are easy and do not include a drying step include PMID: 27038389, PMID: 29204936, PMID 29256082. Likewise, a dry-down step is often included by researchers as a convenience for storage between extraction and MS run, or for cross-sample normalization. Thus, even though many published methods include such a dry-down method, there is no reason to assume that it would not be disposable in prior publications (e.g. PMID: 24631817).

We appreciate this comment and have added additional citations and discussion (page 15, lines 301-307) regarding the existing simple extraction methods of GC-MS using relatively more polar solvents (water/iso-propanol/acetonitrile) (PMID:27038389) and applications to non-biological matrices with large volume of samples and/or extractant with drying and reconstitution (PMID: 29204936 and PMID: 29256082). As now clarified (page 16, lines 314-317), we do not intend to imply that these previously established methods are not worthwhile but rather to provide a stream-lined workflow that supports quantification of known environmental chemicals in biological samples while retaining information for untargeted identification of chemicals to enable population-level studies that require high-throughput analysis with limited sample volume.

2. Reproducibility:

2a. Methods are insufficient to enable full reproducibility with regards to data processing. All XCMS parameters should be provided. I also question whether these parameters were appropriately optimized – 40,000 features seems excessive for serum (even 25,000 after blank removal is still very high).

XCMS R code including the parameters is now provided in supplemental material “Code and example files”. The parameters of XCMS have been optimized based on 1) data extraction of targeted chemicals in authentic standard mixtures (200-300 chemicals per

mixture) and those identified in SRM1957 and SRM1958; and 2) results from IPO package, a published tool for automated optimization of XCMS parameters (Libiseller et al. 2015). The XCMS parameters that generate fewer features (~20,000 before filtering as opposed to ~40,000) caused missing of narrow peaks (e.g. PBDE-153) and thus were not used.

2b. any necessary RAMclustR parameters should be provided

RAMclustR code including the parameters is now provided in supplemental material "Code and example files".

2c. Data availability: "Raw data of this study is in the process of being uploaded to Metabolomics Workbench" is insufficient. Please provide accession number.

The accession number is now provided in the revised main text (page 27, line 582-583).

2d. Please consider providing R code for all analyses and figure generation. While not a requirement, this would facilitate reproducibility.

R code for the computational workflow of GC-MS analyses is now provided in Supplemental material "Code and example files". For figures generated using R, the code is provided in "Source Data". Other figures were generated by SigmaPlot 14.0 (Systat Software, Inc.) and raw data are now provided in "Source Data".

2e. "Starting initially with a QuEChERS procedure (19, 20), we systematically varied solvent composition, volume and extraction time to obtain a simplified procedure with a minimal number of steps and possibilities for contamination and variability in recovery". Presenting that data would facilitate reproducibility, and would also enable other researchers to build on these optimization steps. As currently presented, the rationale for the selection of the optimized XLE method is unknown.

The data using dSPE from QuEChERS are now provided in Supplemental Figure S1. As explained above for Reviewer #2, we did not find difference between QuEChERS-dSPE and high purity MgSO₄ in most targeted chemical analysis. Compared to QuEChERS – dSPE, high purity MgSO₄ showed similar reproducibility in human serum and plasma yet may provide benefit for certain targeted chemicals, such as low-intensity PBB-153.

3. Approach:

3a. Extraction ratios are inconsistent across sample types. The authors should provide a rationale for why these different values were selected rather than a standard ratio based on dry tissue weight.

We appreciate this concern and have corrected the error. In practice, we kept the extraction ratio to be 1:4 to 1:5 for tissue samples (i.e. lung and thyroid) and 1:5 for stool samples (dry weight to solvent volume, mg to μL). The variation of 1:4 from 1:5 in lung extraction was arbitrary in consideration of the lower density of lung as an organ. This information is now added (page 22, lines 469-476).

3b. Using these different extraction ratios likely leads to great variability in terms of chemical concentration and matrix effects across sample types. Thus, comparison across sample types is inappropriate, as is performed in Figure 4C. This issue is compounded by the authors' normalization of tissue data to serum standards: signal intensity and/or standard curve response cannot be assumed to be consistent across matrices. Reference standardization is only appropriate for samples vs references of the same type.

We have revised Figure 4C to remove comparison between plasma and lung. We have also added new data in Figure 3C and Supplemental Table S3 to show comparable and reproducible quantification in the lung based on reference standardization. We have added a discussion on this important issue in comparison across sample types (page 16, lines 334-348).

3c. For best comparability, standards should be prepared in the same solvent as sample extracts, rather than in isoctane.

We have updated the figures (Figure 2C, 2E, 3B and 3C) using standards prepared in the same solvent as sample extracts (hexane and ethyl acetate).

4. Data interpretation and statistics:

4a. y axis in fig 2E is misleading. The larger values for p,p'-DDE make it appear like the measured vs certified values are similar for all the other chemicals, when in fact several are only at ~75% for PCBs.

We have updated the figures using standards prepared in the same solvent as sample extracts (hexane and ethyl acetate). The full comparison of all 40 PCBs, 17 pesticides and 11 PBDEs are updated in Supplemental Table S1.

4b. Likewise, the authors bias the interpretation of this data by only selecting the most successful PCBs from Table S1 to be plotted. Indeed, many are highly divergent (e.g. PCB 149)

As now added to the main text (page 7, lines 141-143), by using the same solvent as for sample preparation, we found that almost all PCBs are detected at 65-130% of the

certified values, which is comparable to other analytical methods. We have now provided data for all 71 chemicals detected in SRM-1958 in Table S1.

4c. Although the authors (rightly) criticize the excessive focus of exposomics studies on a small list of known chemicals, they too focus predominantly on a restricted list of 14-49 compounds in participant samples and only briefly address additional compounds. Section “Analysis of unidentified environmental chemicals” would benefit from additional annotation efforts, and from analysis of collected tissue data.

As now included in Figure 5B and 5D, untargeted analysis of lung data identified three spectra clusters, representing three potential chemicals, which are the same as identified in plasma samples. One of the three had accurate mass corresponding to $C_{27}H_{44}O$ and annotated as cholecalciferol, had high confidence based on n-alkane retention time index which was subsequently confirmed with authentic standard confirmation. These results, now presented in Figure 5, showed that XLE maximizes capture of chemical information that can be later analyzed and identified using an untargeted approach.

4d. Fig 3C: did the authors consider normality? Pearson correlation may not be the most appropriate here.

We appreciate this suggestion. Spearman correlation is now used in this figure.

Minor concerns:

1. Abstract: 200 μ L plasma and 100 mg tissue samples are not small amounts.

We have deleted “small”.

2. Figure 1A is uninformative.

We have removed Figure 1A.

3. Figure 4B data: It is relatively unsurprising that chemicals from the same class would be correlated to each other since they likely derive from similar environmental exposure events.

We agree and have added to the interpretation of this results (page 11, lines 232-234).

4. Typo in table S4 title “intensity”

Corrected.

5. Spearman correlation is used in results but not mentioned in Methods.

We have added it to Methods (page 26, lines 556-558).

6. Line and page numbers would be appreciated in the future.

We have added line and page numbers.

REVIEWER COMMENTS

Reviewer #1 (Remarks to the Author):

The authors have responded to my comments satisfactorily. This paper will be well received by the community.

Reviewer #2 (Remarks to the Author):

The authors have satisfactorily answered the comments raised from the review process.

Reviewer #3 (Remarks to the Author):

Overall, Hu et al have substantially improved their manuscript in response to reviewer comments. However, the following issues were not adequately addressed.

1. Response to 2e is insufficient. While it is good that the authors provided additional data comparing SPE to MgSO₄ treatment in Figure S1, there is still no support for the statement by the authors that they “systematically varied solvent composition, volume and extraction time”. Authors should show one figure with the data for the different solvents they tested, the different volumes, and the different extraction times. This is important to demonstrate that their method is indeed optimized rather than developed by convenience or change.
2. Response to 3a and 3b (extraction ratios) is insufficient. First of all, lines 469 to 470 state that “Other materials were processed similarly with a consistent ratio of 1:5 (sample to hexane-ethyl acetate mixture)”, even though the authors acknowledge that the ratio for the lung is 1:4. Second, even though the ratio of sample to hexane-ethyl acetate was held constant, the relative concentrations of water and formic acid to sample weight is inconsistent. At minimum, authors should rephrase this section of the methods to avoid giving the impression that extraction parameters were held constant, for example by deleting the phrase: “Other materials were processed similarly with a consistent ratio of 1:5 (sample to hexane-ethyl acetate mixture)”.
3. Response to 4c (untargeted analysis): figure 5D match appears to be very poor, with considerable non-matched peaks in the low m/z range. Authors should present an explanation for all the missed peaks in the experimental spectra or omit this match.

4. There is still the typo: "intensity" in a table header (in Table S7)

REVIEWER COMMENTS

Reviewer #1 (Remarks to the Author):

The authors have responded to my comments satisfactorily. This paper will be well received by the community.

Thank you.

Reviewer #2 (Remarks to the Author):

The authors have satisfactorily answered the comments raised from the review process.

Thank you.

Reviewer #3 (Remarks to the Author):

Overall, Hu et al have substantially improved their manuscript in response to reviewer comments. However, the following issues were not adequately addressed.

1. Response to 2e is insufficient. While it is good that the authors provided additional data comparing SPE to MgSO₄ treatment in Figure S1, there is still no support for the statement by the authors that they “systematically varied solvent composition, volume and extraction time”. Authors should show one figure with the data for the different solvents they tested, the different volumes, and the different extraction times. This is important to demonstrate that their method is indeed optimized rather than developed by convenience or change.

We have added data from experimental solvent composition, volume and extraction and presented the results in Supple Figure S1.

2. Response to 3a and 3b (extraction ratios) is insufficient. First of all, lines 469 to 470 state that “Other materials were processed similarly with a consistent ratio of 1:5 (sample to hexane-ethyl acetate mixture)”, even though the authors acknowledge that the ratio for the lung is 1:4. Second, even though the ratio of sample to hexane-ethyl acetate was held constant, the relative concentrations of water and formic acid to sample weight is inconsistent. At minimum, authors should rephrase this section of the methods to avoid giving the impression that extraction parameters were held constant, for example by deleting the phrase: “Other materials were processed similarly with a consistent ratio of 1:5 (sample to hexane-ethyl acetate mixture)”.

We have rephrased and provide more accurate information in this paragraph (page 23, lines 480-484).

3. Response to 4c (untargeted analysis): figure 5D match appears to be very poor, with considerable non-matched peaks in the low m/z range. Authors should present an explanation for all the missed peaks in the experimental spectra or omit this match.

We appreciate this concern. We have updated figure 5D to show matches to library search in the low m/z range which were not clustered into L313 spectra. This was due to lower correlations with L313 spectra across all samples, which may result from possible matrix interference, especially in low m/z range (source data now provided for Fig 5D). We have added an explanation in the figure legend and main text (page 13, lines 264-268).

4. There is still the typo: “intensity” in a table header (in Table S7)

Corrected. Thank you.